# Insulated Switches: Dual-Function Protein RalGEF^RGL-1^ Promotes Developmental Fidelity

**DOI:** 10.3390/ijms21207610

**Published:** 2020-10-15

**Authors:** Tam Duong, Neal R. Rasmussen, David J. Reiner

**Affiliations:** 1Institute of Biosciences and Technology, Texas A&M Health Science Center, Texas A&M University, Houston, TX 77030, USA; tduong@tamu.edu (T.D.); nrasmussen@tamu.edu (N.R.R.); 2Department of Translational Medical Science, College of Medicine, Texas A&M Health Science Center, Texas A&M University, Houston, TX 77030, USA

**Keywords:** bifunctional, toggle, network, PI3K, PI3 Kinase, PTEN, DAF-18 LET-23, LIN-3, DSL

## Abstract

The *C. elegans* vulva is an excellent model for the study of developmental biology and cell–cell signaling. The developmental induction of vulval precursor cells (VPCs) to assume the 3°-3°-2°-1°-2°-3° patterning of cell fates occurs with 99.8% accuracy. During *C. elegans* vulval development, an EGF signal from the anchor cell initiates the activation of Ras^LET-60^ > Raf^LIN-45^ > MEK^MEK-2^ > ERK^MPK-1^ signaling cascade to induce the 1° cell. The presumptive 1° cell signals its two neighboring cells via Notch^LIN-12^ to develop 2° cells. In addition, Ras^LET-60^ switches effectors to RalGEF^RGL-1^ > Ral^RAL-1^ to promote 2° fate. Shin et al. (2019) showed that RalGEF^RGL-1^ is a dual-function protein in VPCs fate patterning. RalGEF^RGL-1^ functions as a scaffold for PDK^PDK-1^ > Akt^AKT-1/2^ modulatory signaling to promote 1° fate in addition to propagating the Ras^LET-60^ modulatory signal through Ral^RAL-1^ to promote 2° fate. The deletion of RalGEF^RGL-1^ increases the frequency of VPC patterning errors 15-fold compared to the wild-type control. We speculate that RalGEF^RGL-1^ represents an “insulated switch”, whereby the promotion of one signaling activity curtails the promotion of the opposing activity. This property might increase the impact of the switch on fidelity more than two separately encoded proteins could. Understanding how developmental fidelity is controlled will help us to better understand the origins of cancer and birth defects, which occur in part due to the misspecification of cell fates.

## 1. Introduction

The *C. elegans* vulva is an excellent model in which to study cell–cell signaling. Six specialized equipotent epithelial cells, the vulval precursor cells (VPCs), are arranged along the ventral midline. During the L3 stage, the VPCs are induced to form the pattern of 3°-3°-2°-1°-2°-3° cell fates. The 1° and 2° cells go through three rounds of division and proceed to form the vulva; an epithelial tube connects the uterus with the external environment. The 3° cells divide once and then fuse with the surrounding epithelium. This pattern occurs with 99.8% accuracy [1,2]. A recent study from Shin et al. (2019) revealed that this developmental fidelity is imposed in part by a signaling protein, RalGEF^RGL-1^, that encodes dual functions [2]. The study concluded that RalGEF^RGL-1^ orchestrates opposing 1°- and 2°-promoting modulatory cascades to help promote developmental fidelity. However, the study hints at a far more important interpretation of bifunctional RalGEF^RGL-1^.

Three models are used to frame our understanding of *C. elegans* VPC fate patterning: (1) the sequential induction model, (2) the graded signal model, and (3) the mutual antagonism model. In the sequential induction model, induction occurs in a stepwise fashion: the 1° fate is induced first, then the 2° fate. The EGF^LIN-3^ ligand, produced by the anchor cell (AC) in the ventral gonad, activates EGFR^LET-23^ in the nearest VPC, which, in turn, activates Ras^LET-60^ to trigger activation of the Raf^LIN-45^ > MEK^MEK-2^ > ERK^MPK-1^ canonical MAP kinase cascade to induce 1° fate [3,4,5] Figure 1). This signal is the core cascade for inducing 1° fate: each component is necessary and sufficient for 1° cell fate. Induced 1° cells synthesize DSL ligands, which via lateral signaling induce the two neighboring VPCs via Notch^LIN-12^ to assume 2° fate [6,7,8]. This second signal is the core cascade for 2° fate: Notch^LIN-12^ is necessary and sufficient to induce 2° fate.

The graded signal model is based on the observation that the position of isolated VPCs within the EGF^LIN-3^ gradient can dictate their ultimate fate [9,10]. Furthermore, variation in the dose of EGF^LIN-3^ or the activation of EGFR^LET-23^ modulates VPC fate selection [11,12]. Taken together, these observations indicate that VPCs are influenced by spatially graded EGF^LIN-3^. However, two lines of evidence suggest that sequential induction—1°, then 2°—takes precedence over the graded signal. First, mutations disrupting either the well-known EGF^LIN-3^ > Ras^LET-60^ > MAP kinase or Notch cascades result in absent 1° or 2° cells, respectively. Second, while disruption of the EGF^LIN-3^ > Ras^LET-60^ > MAP kinase results in a vulvaless phenotype, i.e., no induction of any vulval fates [5], the disruption of Notch^LIN-12^ results in missing 2° cells but the continued presence of 1° cells [7]: 2° cells are never observed in the absence of Notch^LIN-12^ but induction of 1° fate is not compromised. To restate, 1° fate can be induced in the absence of 2°-promoting signal, but 2° cells cannot be induced in the absence of 1°-promoting signal. The sequential induction and graded signal models were not reconciled for a long time [13].

We were able to reconcile these two models by discovering the mechanism by which the graded signal is interpreted: effector switching by Ras^LET-60^ (Figure 1). Oncogenic Ras has been known for decades to use different oncogenic effectors: Raf, PI3K and RalGEF [14,15]. Of these, RalGEF signaling through Ral and further downstream is poorly understood [16]. We found that during the patterning of VPC fates, Ras^LET-60^ switches effectors, from Ras^LET-60^ > Raf^LIN-45^ promoting 1° fate to Ras^LET-60^ > RalGEF^RGL-1^ > Ral^RAL-1^ promoting 2° fate [17]. Ras^LET-60^ signaling through Raf^LIN-45^ is the core signaling cascade for inducing 1° fate. Conversely, Ras^LET-60^ signaling through RalGEF^RGL-1^ > Ral^RAL-1^ is a modulatory signaling cascade that functions in support of the core induction of 2° fate by Notch^LIN-12^. Apparently, the core Ras^LET-60^ > Raf^LIN-45^ (1°-promoting) and Notch^LIN-12^ (2°-promoting) cascades provide the skeleton of VPC patterning, while the graded signaling is overlaid to improve the fidelity of this process [18]. Ral^RAL-1^ propagates this 2°-promoting signal by activation of a MAP4K > p38 MAP kinase signaling cascade [19]. The mechanism by which Ras^LET-60^ switches between utilization of Raf^LIN-45^ and RalGEF^RGL-1^ effectors remains unknown. Mirroring the 2°-promoting Ras^LET-60^ > RalGEF^RGL-1^ > Ral^RAL-1^ modulatory signal in support of Notch^LIN-12^, an PI3K^AGE-1^ > PDK^PDK-1^ > Akt^AKT-1/2^ modulatory signal promotes 1° fate in support of the core Ras^LET-60^ > Raf^LIN-45^ 1°-promoting signal [2,20]. Activation upstream of PI3K^AGE-1^ is unknown: it could be another Ras^LET-60^ input, or it could be EGFR or InsR^DAF-2^. To summarize, to properly pattern the VPCs at high fidelity, each vulval contributing cell fate type—1° and 2°—requires a core, essential signaling cascade and a modulatory, inessential signaling cascade.

We have also postulated the mutual antagonism model [21]. In response to initial patterning of VPC fates by the two core signaling cascades, Ras-Raf-MAP kinase and Notch, a series of additional signals and transcriptional changes in signaling molecules occur. Collectively, these changes appear to reinforce initial patterning events and/or reduce conflicting signals within cells. Perhaps VPCs thereby avoid aberrant or ambiguous cells fates that would compromise the function of the vulva and hence decrease reproductive fitness. For example, in presumptive 1° cells, the Notch^LIN-12^ receptor is internalized and degraded, thus preventing conflicting 2°-promoting signals in a VPC that is assuming 1° fate [22,23]. Conversely, presumptive 2° cells express ERK^LIP-1^ phosphatase (DUSP or MAP kinase phosphatase “MKP” in Figure 1) as a Notch^LIN-12^ transcriptional client gene to quench inappropriate ERK^MPK-1^ activation in these cells, thus preventing conflicting 1°-promoting signals in VPCs that are assuming 2° fate [24]. A series of other regulatory events presumably contribute similarly to preventing contradictory signals, and thus are likely to increase the developmental fidelity of the system (reviewed in [21]).

Here, we reflect on a study that suggests an additional, fourth model that increases developmental fidelity, which we term “insulated switches” in signaling. Specifically, Shin et al. (2019) show that i) RalGEF^RGL-1^ is a dual-function protein in VPC fate patterning, ii) that the two functions are genetically separable, iii) that RalGEF^RGL-1^ likely functions as a scaffold for PDK^PDK-1^ > Akt^AKT-1/2^ signaling in addition to propagating the Ras^LET-60^ signal through Ral^RAL-1^, and iv) that deletion of RalGEF^RGL-1^ does not alter the relative balance of 1° and 2° fates in VPC patterning but v) increases the frequency of patterning errors 15-fold about the wild-type control [2]. In other words, RalGEF^RGL-1^ orchestrates the two opposing modulatory signals in VPC fate patterning: Akt promotion of 1° fate and Ral promotion of 2° fate. That both activities are embodied in the same signaling molecule raises thought-provoking questions about the functional significance of this concurrence. We speculate that both signals embodied in the same protein represent an “insulated switch,” whereby the promotion of one signaling activity curtails the promotion of the opposing activity and vice-versa, perhaps thus increasing the impact of the switch on fidelity more than could two separately encoded proteins. This model also casts the frequently described phenomenon of “pathway cross-talk” in an interesting new light.

## 2. Two Antagonistic Functions of RalGEF^RGL-1^

The dual function of RalGEF^RGL-1^ was uncovered serendipitously: the deletion of Ral^RAL-1^ reduced 2°-promoting signaling while the deletion of RalGEF^RGL-1^ caused no effect on patterning [2]. These results were paradoxical. Since RalGEF^RGL-1^ functions as an intermediary between Ras^LET-60^ and Ral^RAL-1^ [25], we expect the deletion of either RalGEF^RGL-1^ or Ral^RAL-1^ to cause the same signaling defect. A natural hypothesis, eventually confirmed, was that RalGEF^RGL-1^ plays opposing and genetically separable roles in VPC fate patterning. Answering this question was technically demanding. Since the VPCs induced at the mid-L3 stage comprise approximately one-ten thousandth of the volume of the animal, relatively minute and transient changes in conventional signaling biomarkers like phospho-ERK or phospho-Akt are not detectable. Furthermore, no transcriptional reporters have been described as readouts of the modulatory cascades of Ras^LET-60^ > RalGEF^RGL-1^ > Ral^RAL-1^ or PI3K^AGE-1^ > PDK^PDK-1^ > Akt^AKT-1/2^ during patterning of VPC fates, precluding a quantitative readout of fluorescent reporters of activity of those cascades. Consequently, the authors used phenotypic analysis to dissect these signals, by counting VPCs whose fate were transformed from 3° to 1° or 3° to 2° [2,17].

To understand these approaches, we need to understand the use of sensitized genetic backgrounds to reveal pathway activity through the use of genetic principles of parallelism and epistasis [26,27,28]. Single mutations abrogating the function of RalGEF^RGL-1^ or Ral^RAL-1^ conferred no obvious VPC patterning defects in an otherwise wild-type animal, consistent with a modulatory signal rather than the necessary and sufficient core signals [17,19]. Shin et al. (2019) used a constitutively activating mutation in Ras^LET-60^, *let-60(n1046*gf*)*, which harbors a moderately activating G13E mutation, to provide a “sensitized” genetic background that responds to genetic perturbations of modulatory proteins. *let-60(n1046*gf*)* confers 3°-to-1° cell fate transformations, resulting in ectopic 1° lineages (also called “multivulva,” or “Muv”). Mutations that reduce the function of genes that act downstream of Ras^LET-60^ suppress the induction of *let-60(n1046*gf*)*-dependent ectopic 1° cells (e.g., [29,30]). Mutations that increase the activity of the PI3K^AGE-1^ > PDK^PDK-1^ > Akt^AKT-1/2^ cascade increase the frequency of *let-60(n1046*gf*)*-dependent ectopic 1° cells, while a reduction in this cascade function reduces induction of ectopic 1° cells [20]. In contrast, based on the principle of mutual antagonism – 2° signals are antagonistic to 1° signals and vice versa – a reduction in 2°-promoting signals increases the induction of ectopic 1° lineages [2,17]. Such analyses take advantage of the principles of genetic parallelism: the genetic perturbation of cascades that function in parallel to *let-60(n1046*gf*)* 1°-promoting activity partially enhance or suppress the induction of ectopic 1° cells, as assayed by counting induced cells [2,17,19].

Shin et al. (2019) were able to unravel this signaling puzzle using these concepts of parallelism and epistasis. First, the authors defined genetic tools to selectively ablate the 2°-promoting signal of RalGEF^RGL-1^ > Ral^RAL-1^ without altering the other properties of these proteins. The R139H missense mutation in Ral^RAL-1^ and R361Q putative GEF dead mutation in RalGEF^RGL-1^ selectively disrupt 2° signaling in the VPCs and enhance 1° induction in the *let-60(n1046*gf*)* background [2,19]. These results indicate that RalGEF^RGL-1^ promotes 2° fate via the canonical GEF-dependent activation of Ral^RAL-1^. In an experiment that bypassed the GEF requirement of RalGEF^RGL-1^ with activated Ral, the effect of deletion of RalGEF^RGL-1^ on the 1° promoting function could be detected. Therefore, RalGEF^RGL-1^ performs an additional GEF-independent function that antagonizes its canonical function.

To further investigate the non-canonical GEF-independent activity of RalGEF^RGL-1^ in promoting 1° fate, epistasis experiments were performed that point to the deletion of RalGEF^RGL-1^ suppressing the effect of deletion PTEN^DAF-18^ and constitutively activated PDK^PDK-1^, and partially suppressing the effect of constitutively activated Akt^AKT-1^. These results are consistent with the model that RalGEF^RGL-1^ functions as a scaffold for PDK^PDK-1^ and AKT^AKT-1^ in the modulatory 1°-promoting PI3K^AGE-1^ cascade. This observation is consistent with mammalian studies showing that mammalian RalGDS physically interacts with PDK and Akt as a scaffold [31,32]. However, these observations are also consistent with RalGEF^RGL-1^ functioning in parallel to PI3K^AGE-1^ > PDK^PDK-1^ > Akt^AKT-1^. Thus, RalGEF^RGL-1^ likely balances the functions of two modulatory cascades: canonical GEF-dependent activation of Ral^RAL-1^ and promotion of 2° fate, and non-canonical GEF-independent scaffolding of PDK^PDK-1^ and Akt^AKT-1^ to promote 1° fate.

To circumvent the requirement for a sensitized background, the authors counted patterning errors in the wild type vs. two strains with deleted RalGEF^RGL-1^, which requires the scoring of many more animals than used in sensitized backgrounds (1200 per genotype). Strikingly, the baseline error rate in VPC patterning was 15-fold higher in the two deletion mutants than in the wild type (*P* < 0.0001). Notably, environmental perturbations had no effect on different mutant backgrounds [2]. These observations indicate that the RalGEF^RGL-1^ function is not to defend against environmental variability but to decrease the variability in the system, i.e., fluctuations in signaling output or “noise”.

The rate of patterning errors changed from 0.2% in the wild type to 3.0% upon loss of both RalGEF^RGL-1^ functions. Most animals still patterned the VPCs correctly using the core signaling cascades of Ras^LET-60^ > Ral^RAL-1^ > MAP kinase and Notch^LIN-12^. However, this 15-fold change in rate of developmental error reveals the function performed by modulatory cascades in patterning VPCs fates. These cascades likely reinforce initial commitment to fates analogously to the mechanisms of mutual antagonism discussed above. If both activities were encoded separately, a GEF intermediate between Ras^LET-60^ and Ral^RAL-1^ on the one hand and a scaffold for PDK^PDK-1^ on the other, we would never take note of the phenomenon. It is the dual function of RalGEF^RGL-1^ in the process that draws our attention as unusual. Does the bifunctionality of RalGEF^RGL-1^ have special mechanistic significance? We next discuss the possible meanings of this phenomenon.

## 3. Speculative Model: the Dual Function RalGEF^RGL-1^ Works as An “Insulated Switch”

We are intrigued by the observation that opposing functions are encoded in the same protein. How is such a phenomenon different mechanistically from the same activities being encoded in two separate proteins, e.g., a signal transducer and a scaffold? In simple linear views of signaling, the phenomenon of the bifunctional protein would not matter. However, we speculate that the observed mechanisms are not a meaningless coincidence. We hypothesize that RalGEF^RGL-1^ is an “insulated switch”: both activities in the same protein are important to reduce signaling noise (Figure 2).

We envision the insulated switch being a mechanism whereby the activity promoting one signal antagonizes the other signal, and vice versa. We foresee two potential mechanisms by which an insulated switch could reduce signaling noise: (1) cell biological or (2) steric hindrance. In a cell biological mechanism, the putative RalGEF^RGL-1^ insulated switch is recruited to discrete subcellular compartments for each activity that does not support contribution to the opposing activity. For example, recruitment of RalGEF^RGL-1^ to the plasma membrane by upstream activated Ras^LET-60^ may sequester RalGEF^RGL-1^ away from the subcellular address in which RalGEF^RGL-1^ would function as a scaffold for PDK^PDK-1^ and Akt^AKT-1^, while scaffolding would sequester away from potential activation by Ras^LET-60^. Conversely, in a steric hindrance or physical interference model, Ras^LET-60^ binding to RalGEF^RGL-1^ would interfere with scaffolding, while scaffolding blocks RalGEF^RGL-1^ accessibility to Ras^LET-60^, to thereby activate Ral^RAL-1^. In the insulated switch model, we imagine that both activities in the same protein increase the fidelity of VPC fate patterning, compared to a situation in which the same activities were encoded in different proteins.

In molecular pathways we tend to think linearly; linearity is built into the word “pathway.” For instance, we tend to think about RalGEF^RGL-1^ as only a GEF of Ral^RAL-1^ and a scaffold is only a scaffold. Yet many larger proteins perform multiple functions. They bind to targets other than those involved in their canonical function. Canonical functions are usually identified accidentally: what did we find first? It is hard to break out of that chronological bias, which leads to a conceptual bias.

These ideas lead us to the concept of “cross talk” in signal transduction. Researchers frequently describe one signaling axis communicating with another signaling axis, and this phenomenon is frequently referred to as “cross talk.” However, many such events are orphans, or stand-alone observations that are not followed up on to investigate the biological consequences of the activity. However, this phenomenon is frequently described in the context of the “canonical pathway” vs the novel cross-talk function, which represents an intrinsic bias. We see this in the naming of RalGEF^RGL-1^ itself: had it been originally discovered as a scaffold for PDK^PDK-1^ and Akt^AKT-1^ signaling, its name would be very different than “RalGEF^RGL-1^”. If we think about these signaling systems as signaling networks instead of pathways, then these dual-functioning proteins are not unusual, but instead a key feature of signaling. Mechanistically, this dual-functional property of proteins at the network level could encode far more richness into signaling networks. Combinatorial functionality produces far more complexity. How do we generate, or build, one of the most complicated animals on the planet with the same number of genes as we use to generate one of the simplest animals? The answer probably is combinatorial functions of proteins: dual functionality (or more than two: many). Importantly, we also can get increased developmental fidelity with this special feature of the dual-functioning protein.

We speculate about the role of the insulated switch: can a dual-functional protein reduce noise more than two single-function proteins, i.e., a signaler and a scaffold? Can the same protein simultaneously promote one activity while also working to reduce the opposing activity? Figure 2 shows the pushmi-pullyu from the children’s story Dr. Doolittle, which serves as a metaphor for the RalGEF^RGL-1^ protein that can function to promote both a primary and secondary fate. How would we test this? Each activity would need to be separately mutated by creating two different genes in the animal so that they are no longer linked. One is only a scaffold, one only a signaler, but both are expressed at same levels and in same cells. This approach might break the insulated signal in RalGEF^RGL-1^, so the signaling activities are still encoded but the switch no longer exists. Since we do not know where the scaffolding activity is encoded in RalGEF^RGL-1^, this approach might not be technically feasible.

We further speculate that proteins in the signaling world that can be identified as obviously multifunctional could be insulated switches. One such protein is PLC-epsilon, an isozyme of the phospholipase C (PLC) family that functions both in the transduction of signal and as a shared effector protein in Ras-, Rho-, and heterotrimeric G protein -mediated signaling. Thus, crosstalk between heterotrimeric and small GTPase signaling pathways is sensed and mediated by multifunctional protein PLC-epsilon (reviewed in [33]). Another interesting example is the SOS protein, which provides dual GEF catalytic activity for both Ras and the small GTPase Rac, a member of the Rho branch of the Ras superfamily. The SOS protein contains Dbl homology (DH, also called RhoGEF) and Pleckstrin homology (PH) domains. The tandem DH-PH domain cluster comprises the majority of GEFs for Rho family GTPases. The catalytic DH domain mediates the GDP/GTP exchange on Rho GTPases, while the PH domain modulates the activity of the DH domain (e.g., promoting membrane association, facilitating GTPase subtract binding, or control intramolecular interactions; reviewed in [34])

## 4. Conclusions

The patterning of the VPC fate in *C. elegans* occurs with 99.8% accuracy. Many layers of signaling network orchestrate together to generate high fidelity by minimizing mis-specified or ambiguous fates. This accuracy is generated by the combination of (1) sequential induction, (2) graded signal, (3) mutual antagonism, and (4) the orchestration of two modulatory cascades promoting 1° and 2° signaling by the dual functioning protein RalGEF^RGL-1^. The study by Shin et al. (2019) revealed this new, latter mechanism by using genetic manipulations, the combined principles of epistasis and parallelism. Cancer often develops from increased noise and error rate, misspecification or variable specification of cells. Birth defects may also be initiated from increased noise during development. Therefore, the signaling networks that reinforce the fidelity of cell fate patterning will provide us with important clues to understand the disease’s origins. For organisms with thirty trillion cells, more noise reduction becomes increasingly important. In mammals, there are four RalGEF/RalGDS-encoding genes. Would these operate as insulated switches with four-fold redundancy? There might be more combinatorial properties, such as scaffold specificity for different PDK and Akt isoforms or splice variants. We conclude that insulated switches could provide additional layers of fidelity and buffering of signaling networks against stochasticity during development.

## Figures and Tables

**Figure 1 ijms-21-07610-f001:**
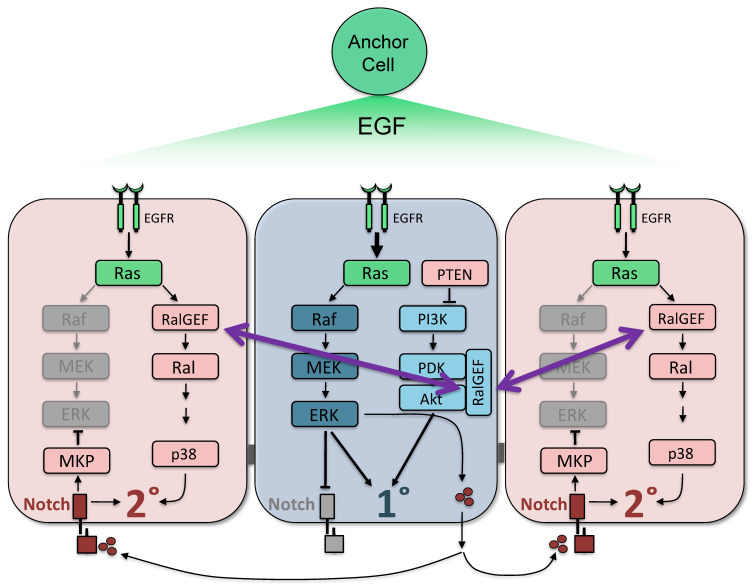
Models of *C. elegans* vulval precursor cells (VPCs) fate patterning and the dual function of RalGEF^RGL-1^ in modulatory signaling cascades. Equipotent VPCs are patterned by graded EGF^LIN-3^ (Morphogen) from the anchor cells (AC). In response to EGF^LIN-3^, EGFR^LET-23^ activates the Ras^LET-60^ > Raf^LIN-45^ > MEK^MEK-2^ > ERK^MPK-1^ canonical MAP kinase signaling cascade to promote 1° fate. The induced presumptive 1° cell synthesizes redundant Notch ligands to laterally signal Notch^LIN-12^ activation to induce neighboring cells to assume 2° fate. In the presumptive 1° cell, activation of EGFR^LET-23^ causes internalization and degradation of Notch^LIN-12^. In the presumptive 2° cells, Notch^LIN-12^ transcriptional target *lip-1* is expressed. *lip-1* encodes ERK phosphatase (MKP) to quench ERK activation in presumptive 2° cells. During vulva fate patterning, Ras^LET-60^ switches effectors from canonical Raf^LIN-45^ to non-canonical RalGEF^RGL-1^ > Ral^RAL-1^ that promotes 2° fate. The dual functions of RalGEF^RGL-1^ are shown with purple arrows, indicating that RalGEF^RGL-1^ promotes both 1° and 2° fate via two modulatory signaling cascades: (1) RalGEF^RGL-1^ promote 1° fate via a non-canonical, GEF-independent activity (scaffold for PDK^PDK-1^ and Akt^AKT-1^ in the modulatory 1°-promoting PI3K^AGE-1^ cascade) (light blue); and (2) RalGEF^RGL-1^ promote 2° fate via canonical GEF-dependent activation of Ral^RAL-1^ (light rose). Necessary and sufficient cascades are in darker blue and darker maroon colors.

**Figure 2 ijms-21-07610-f002:**
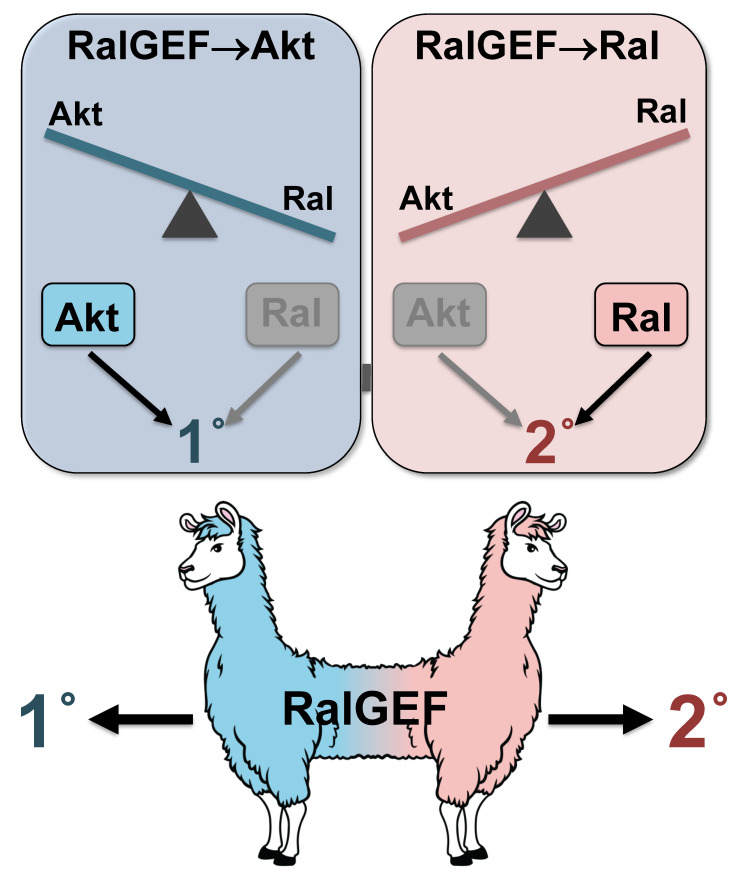
Insulated switches of dual function protein RalGEF^RGL-1^ during VPCs fate patterning. RalGEF^RGL-1^ functions in two modulatory cascades to promote both 1° and 2° fate. RalGEF^RGL-1^ is a scaffold for PDK^PDK-1^ and Akt^AKT-1^ in the modulatory 1°-promoting PI3K^AGE-1^ cascade (light blue). And RalGEF^RGL-1^ promotes 2° fate via canonical GEF-dependent activation of Ral^RAL-1^ (light rose). A pushmi-pullyu animal is used as a metaphor for RalGEF^RGL-1^ protein that has dual function in promoting both 1° and 2° fate.

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
