# Peer review of "Insulated Switches: Dual-Function Protein RalGEF^RGL-1^ Promotes Developmental Fidelity"

_ijms, 2020, doi:10.3390/ijms21207610_

Round 1
Reviewer 1 Report
The Perspective by Duong, Rasmussen and Reiner provides a detailed and interesting discussion on a recent study by the same group, published last year in PLoS Genetics. Reiner and colleagues discovered that RalGEF has dual functions in vulval induction in the nematode Caenorhabditis elegans: it promotes primary cell fate (in the P6.p vulva precursor cell) in a non-canonical GEF-independent manner and it promotes secondary cell fate (in P5.p and P7.p) through GEF-dependent activation of Ral. In the current perspective the authors describe the experimental logic behind the study and propose that RalGEF acts as an “insulated switch” to increase fidelity in vulval patterning. The concept of such dual-acting insulator characteristics contained within a single protein is interesting and the authors argue that it might be a more wide-spread phenomenon that has escaped attention due to the technical difficulties in its detection.
I recommend publication of the manuscript and I have only three very minor suggestions:
Lines 64-65: “Necessary and sufficient cascades are in dark color.” Except for Raf->MEK->ERK, it is not obvious which one(s) the authors refer to.
Lines 80-81: “Of the three oncogenic effectors of Ras/LET-60, Raf/LIN-45, PI3K/AGE-1, and RalGEF/RGL-1> Ral/RAL-1,…”. I would simplify this sentence to “Of the three oncogenic effectors of Ras/LET-60, Raf/LIN-45, PI3K/AGE-1, and Ral/RAL-1,…”.
Line 205: “promote” should be “promotes”.
Author Response
Thank you for the review of our manuscript submitted to the International Journal of Molecular Sciences for your special issue, "Dual Function Molecules and Processes in Cell Fate Decision." We appreciate these inputs. Thus, we are resubmitting our manuscript titled “Insulated switches: Dual function protein RalGEFRGL-1 promotes developmental fidelity.” We have incorporated all of your minor editing concerns in the revised manuscript, indicated as green text.
-Lines 64-65: “Necessary and sufficient cascades are in dark color.” Except for Raf->MEK->ERK, it is not obvious which one(s) the authors refer to.
Our response: In particular, we adjusted the formatting in Figure 1 as requested, to show that Notch is one of the necessary and sufficient signals. The schematic of the receptor is now thicker and the word “Notch” is in larger font, bolded, and color-coded the same as the receptor schematic. We think this will help interpretation of the figure.
-Lines 80-81: “Of the three oncogenic effectors of Ras/LET-60, Raf/LIN-45, PI3K/AGE-1, and RalGEF/RGL-1> Ral/RAL-1,…”. I would simplify this sentence to “Of the three oncogenic effectors of Ras/LET-60, Raf/LIN-45, PI3K/AGE-1, and Ral/RAL-1,…”.
Our response: We edited the sentence to improve the clarity.
Line 205: “promote” should be “promotes”.
Our response: Yes, we fixed it.
Reviewer 2 Report
The vulval cell patterning system in C. elegans has been a long-standing model for signaling, patterning, and even developmental system drift. In this perspective, the authors summarize old and new work in this area and propose how competing activities cause robust assignment of cell fates. The authors describe three models for VPC fate specification and discuss the arguments for and against these. In particular, prior work from this lab has shown that effector switching downstream of Let-60/RAS reconciles the graded signal model with the sequential induction model. Their work leads to the notion that there is an essential core signaling pathway and a modulatory signaling cascade. The authors then describe their elegant (and sophisticated) approach to dissecting the relative contributions of Raf and RalGEF that took advantage of unique missense mutations and the genetics approaches possible in the C. elegans system. Through sensitized backgrounds they were able to reveal the functions of RalGEF. They propose an 'insulated switch' model, and then the paper discusses general features of the genetic dissection of signaling pathways that might have biased the interpretation of such pathways as strictly linear when there is deeper nuance.
This is a timely perspective article that helps to explain the nuance behind the signaling pathways in a specific event (VPC specification in C. elegans) while also attempting to generalize the concept of insulated switches. I recommend acceptance with only minor comments, below.
Minor comments
line 16 – 'EGF signal' – 'an EGF signal' and 'anchor cell' – 'the anchor cell'
line 162 – 'RalGEF(RGL-1) promote' – promotes
line 200 – 'is important to' – are important to
line 244-246 – the parenthetical sentence about Figure 2 can be a sentence separate from the one before it.
line 250 – 'don't' – do not
Author Response
Thank you for the review of our manuscript submitted to the International Journal of Molecular Sciences for your special issue, "Dual Function Molecules and Processes in Cell Fate Decision." We appreciate these inputs. Thus, we are resubmitting our manuscript titled “Insulated switches: Dual function protein RalGEFRGL-1 promotes developmental fidelity.” We have incorporated all of your minor editing concerns in the revised manuscript, indicated as green text.
In particular, we fixed all of the points mentioned below in the revised manuscript.
line 16 – 'EGF signal' – 'an EGF signal' and 'anchor cell' – 'the anchor cell'
line 162 – 'RalGEF(RGL-1) promote' – promotes
line 200 – 'is important to' – are important to
line 244-246 – the parenthetical sentence about Figure 2 can be a sentence separate from the one before it.
line 250 – 'don't' – do not